

# The mechanism and detection of alternative splicing events in circular RNAs

Xiaohan Li, Bing Zhang, Fuyu Li, Kequan Yu and Yunfei Bai

State Key Laboratory of Bioelectronics, School of Biological Sciences and Medical Engineering, Southeast University, Nanjing, Jiangsu, China

## ABSTRACT

Circular RNAs (circRNAs) are considered as functional biomolecules with tissue/development-specific expression patterns. Generally, a single gene may generate multiple circRNA variants by alternative splicing, which contain different combinations of exons and/or introns. Due to the low abundance of circRNAs as well as overlapped with their linear counterparts, circRNA enrichment protocol is needed prior to sequencing. Compared with numerous algorithms, which use back-splicing reads for detection and functional characterization of circRNAs, original bioinformatic analyzing tools have been developed to large-scale determination of full-length circRNAs and accurate quantification. This review provides insights into the complexity of circRNA biogenesis and surveys the recent progresses in the experimental and bioinformatic methodologies that focus on accurately full-length circRNAs identification.

## INTRODUCTION

Circular RNAs (circRNAs), without 5′-3′polarity and polyadenylated (poly(A)) tails, were first discovered in RNA viruses 40 years ago (*Sanger et al., 1976*). CircRNAs are functionally important for these viruses to generate multiple genomic copies by rolling circle amplification of the RNA genome with host DNA-dependent RNA polymerases. Later, a small number of additional endogenous circRNAs were revealed in unicellular eukaryotes (*Grabowski, Zaug & Cech, 1981*), archaea (*Kjems & Garrett, 1988*) and higher eukaryotes (*Capel et al., 1993*; *Cocquerelle et al., 1993*). Whereas, it was not until the raise of Next-generation sequencing (NGS) technology and bioinformatics that numerous circRNAs were discovered in highly diverged eukaryotic organisms (*Guo et al., 2014*; *Memczak et al., 2013*; *Rahimi et al., 2019*; *Westholm et al., 2014*).

Studies have suggested a cell type/tissue- specific manner of circRNAs expression, and some of them were expressed across different species (*Guo et al., 2014*; *Jeck et al., 2013*; *Memczak et al., 2013*; *Salzman et al., 2013*). Based on their origin, circRNAs can be grouped into four categories: exonic circRNAs, only consisted of exons; intronic circRNAs, only consisted of introns; exonic-intronic circRNAs, consisted of exons and introns; and intergenic circRNAs from intergenic regions (*Wang, Nazarali & Ji, 2016*). Unlike linear RNAs, circRNAs show long half-lives and play various biological roles, such as function as

Corresponding author
Yunfei Bai, whitecf@seu.edu.cn

microRNA (miRNA) sponges (*Hansen et al., 2013*), regulating parental gene transcription (*Li et al., 2015*) and cell proliferation (*Bachmayr-Heyda et al., 2015*), interacting with RNA-binding proteins (RBPs) (*Li et al., 2017*), as well as translating proteins (*Meng et al., 2017*). Previous bioinformatics tools have been raised for large-scale circRNAs identification by using back-splicing junction sites (BSJs) to represent different circRNAs (*Gao, Wang & Zhao, 2015*; *Guo et al., 2014*; *Zhang et al., 2014*). However, the functional and evolutionary analyses of circRNAs depends on their full-length sequences. Considering the prevalence of alternative circRNAs processing, such as exon skipping and intron retention (*Dong et al., 2017*; *Gao et al., 2016*; *You et al., 2015*; *Zhang et al., 2016a*), the aforementioned methods may provide inaccurate information of circRNA isoforms which have same BSJs but differ in their internal compositions. To solve the above problems, much effort has been made for accurate determination of full-length circRNAs. In this review, we summarize the recent findings on biogenesis of circRNA isoforms, progress on detection methods, and highlight the challenges for further research.

## SURVEY METHODOLOGY

Article searching was performed in Web of Science, PubMed and Elsevier with the words ''alternative splicing'', ''alternative back-splicing'' or ''circRNAs enrichment'' in combination with the terms ''circRNAs'' or ''full-length circRNAs'' in the title and abstract. All studies published both in English and in Chinese were searched with no restriction on the publication period. In addition, the reference lists of the retrieved articles were manually searched to identify potentially relevant studies. We focused mainly on the biogenesis of circRNAs, factors resulting in alternative splicing and experimental and bioinformatic methodologies which are expected to accurately discover full-length circRNAs.

### Biogenesis of CircRNAs

Recent studies unveiled that circRNAs are not only derived from numerous of precursor mRNAs (pre-mRNAs) but also long non-coding RNAs (lncRNAs) (*Burd et al., 2010*; *Holdt et al., 2016*; *Huang et al., 2018*; *Zhang et al., 2019*). Two mechanisms have been raised to explain the formation of circRNAs (Fig. 1), which are defined as direct back-splicing and lariat intermediate (exon skipping) (*Jeck & Sharpless, 2014*). Broadly speaking, both models involve back-splicing and canonical splicing, but they are notably different in which step happens first (*Chen & Yang, 2015*).

#### *Direct back-splicing*
Complementary sequences across long flanking introns (*Liang & Wilusz, 2014*; *Zhang et al., 2014*) or the dimerization of RBPs (*Ashwal-Fluss et al., 2014*; *Conn et al., 2015*; *Li et al., 2017*) can facilitate 'direct back-splicing' by bridging the splice donor site into close proximity to the acceptor site (Fig. 1A) (*Chen & Yang, 2015*). Therefore, the branch point upstream of a circularized exon is able to attack the downstream splice donor site, resulting in a pre-mRNA intermediate containing a $2'$, $5'$-phosphodiester linkage. Subsequently, the upstream splice acceptor is attacked by the free $3'$ hydroxyl group of the prospectively circRNA to form a circRNA. It is worthwhile noting that one set of double-stranded

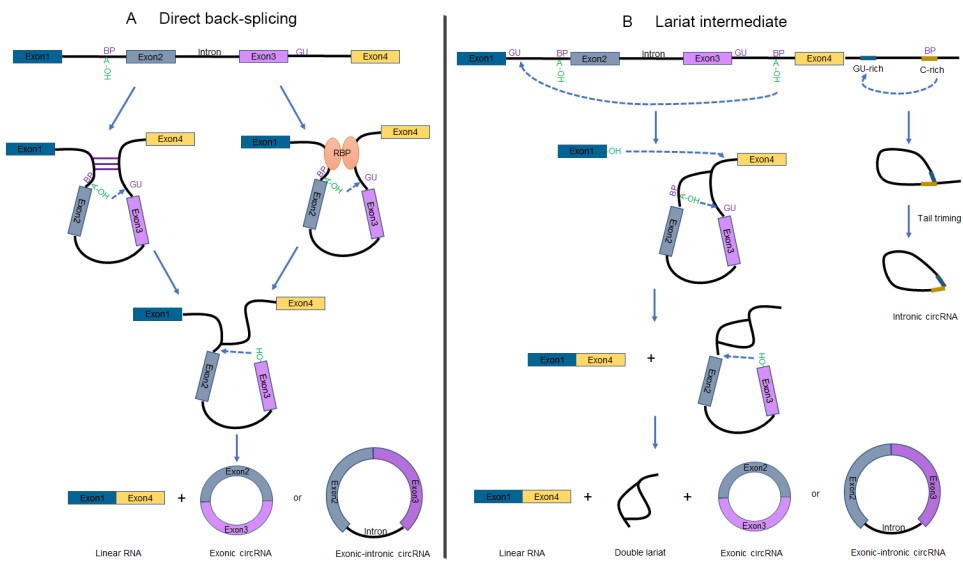

**Figure 1** **Schematic of circRNA formation.** (A) The "direct back-splicing" model for the generation of circRNAs. The loop structure can be formed by either complementary sequences across long flanking introns (left) or RBPs (right). The intron sequences are removed or retained to generate Exonic circRNAs or Exonic-intronic circRNAs, respectively. (B) The "lariat intermediate" model for the generation of circRNAs. In this model, canonical splicing appears first, forming a lariat precursor. And then internal back-splicing take place, which result in a double lariat molecular and a circRNA (left). The generation of Intronic circRNAs relies on GU-rich sequences and C-rich sequences to form a lariat intron, followed by tail triming (right). **BP,** branchpoint; **RBPs,** RNA binding proteins.

RBPs, Adenosine deaminase acting on RNA (ADARs), suppress circRNA expression by destabilizing intron pairing interactions (*Ivanov et al., 2015*).

### *Lariat intermediate*

Interestingly, inverted repeated elements flanking the circle-forming exons are widespread in mammals but rare in lower eukaryotes. Lariat intermediate mechanism has been proposed in the literature accounting for producing circRNAs (Fig. 1B, left). Using fission yeast (*Schizosaccharomyces pombe*) as a model system, researchers found that mrps16 gene produced a circRNA by forming an exon-containing lariat firstly (*Barrett, Wang & Salzman, 2015*). In this model, canonical splicing appears first, forming a lariat precursor (*Eger et al., 2018*). And then internal back-splicing take place, which result in a double lariat molecular and a circRNA.

According to this approach, alternative exons or introns are excised from the pre-mRNA as exon-intron(s)-exon intermediate molecules. Lariat structure is not permanently stable for the reason that the 2′ to 5′ phosphodiester bond will be recognized and debranched specifically to linear form by debranching endonucleases (*Hesselberth, 2013*). However, some lariats are relatively stable and may probably escape from debranching to forming intronic circRNAs (ciRNAs), when contain consensus RNA sequences (a 7-nt GU-rich sequence near the 5′ splice site and an 11-nt C-rich sequence near the branch point). These

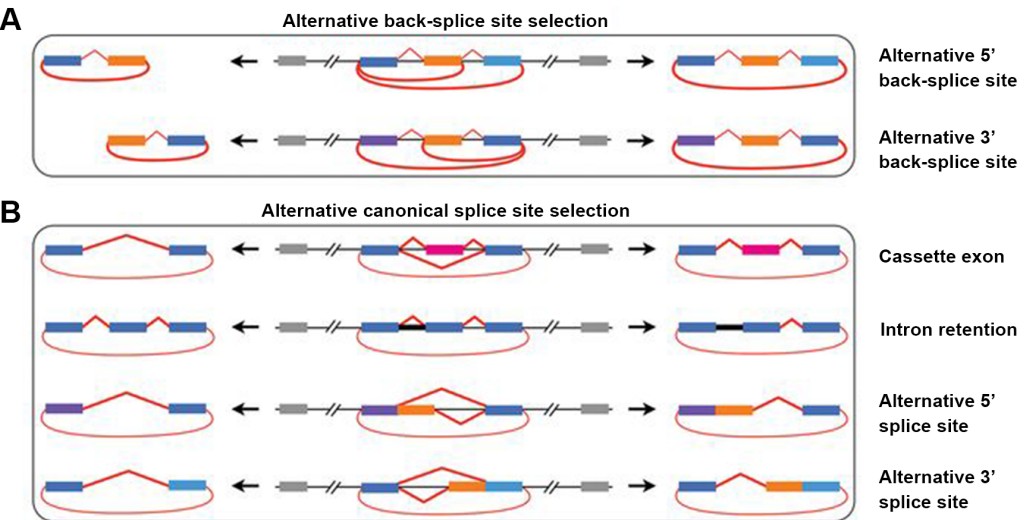

**Figure 2** **Schematic diagrams of alternative (back-) splicing** (*Zhang et al., 2016a*). (A) Two types of alternative back-splicing. (B) Four basic types of alternative canonical splicing. Colored bars: exons; black lines: introns; red polylines: (canonical) collinear splicing; red arc lines: back-splicing (circularization). Compared with the original image, the font color was changed to black.

ciRNAs are distinguished from exonic circRNAs by a $2'$ to $5'$ phosphodiester bond (Fig. 1B, right) (*Lasda & Parker, 2014*; *Zhang et al., 2013*).

## Alternative splicing within CircRNAs

Alternative splicing (AS) in protein-coding or non-coding genes has a dramatic impact on in cellular differentiation and organismal development. Similar to linear isoforms, circRNAs generated from multi-exon genes are alternatively spliced as well, and may further plays important roles in the transcriptome (*Feng et al., 2019*; *Gao et al., 2016*). AS events in circRNAs fall into two categories: circRNAs with alternative BSJs (ABSJs) and circRNA with same BSJs but distinct internal compositions (Fig. 2).

As for ABSJs, a single gene could produce multiple circRNAs which contain different $5'$ splice donors or $3'$ splice acceptors (Fig. 2A) (*Zhang et al., 2016a*). For instance, DNMT3B and XPO1 gene in human produced multiple highly expressed circRNAs through this mechanism. Experimental methods, such as reverse transcription-PCR (RT-PCR) or RNA-sequencing (RNA-seq), are able to validate this phenomenon (*Beltrán-García et al., 2020*; *Jeck & Sharpless, 2014*).

Different from ABSJs, the occurrence of alternative canonical splicing within circRNAs leads to isoforms that have the same BSJs but distinct in internal compositions (*Gao et al., 2016*; *Zhang et al., 2016a*). CircRNAs derived in this manner, can be grouped into one of four categories: cassette exon, intron retention, alternative $5'$ splicing, and alternative $3'$ splicing (Fig. 2B) (*Gao et al., 2016*; *Jeck et al., 2013*; *Memczak et al., 2013*; *Salzman et al., 2013*; *Zhang et al., 2016a*; *Zhang et al., 2014*). Researchers identified that numerous new exons retained in circRNAs by specific alternative canonical splicing. Consistent with this, several previously unannotated exons in the human MED13L locus were discovered by

analyzing multiple poly(A)- RNA-seq (*Zhang et al., 2016a*). Additionally, circRNAs with or without a retained intron generated from CAMSAP1 locus, were confirmed by northern blotting assay (*Salzman et al., 2013*; *Zhang et al., 2014*). Similarly, a cassette exon in the human XPO1 locus was proved to be circRNA-predominant (*Zhang et al., 2014*).

Previous researches showed that more than 50% of the expressed loci could produce two or more circRNA isoforms by AS (*Gao et al., 2016*; *Ji et al., 2019*; *Ottesen et al., 2019*; *Rui et al., 2018*; *Zhang et al., 2016a*; *Zheng et al., 2019*). According to Feng et al., AS events in circRNAs were potentially involved in cancer processes. For instance, in UBAP2L locus, alternative 5′ splicing occurred in circRNAs between cancer and adjacent normal tissues (*Feng et al., 2019*). Similarly, skipping exon event occurred in circRNAs from RAB6A. Therefore, it is important to explore circRNA AS events, and ongoing investigations have begun to focus on factors which were linked to AS, such as circRNA-Rolling Circle Amplification and CIRI-full (*Das et al., 2019*; *Zheng et al., 2019*).

## The mechanisms of alternative splicing events

Since circRNA processing is related to transcription and pre-mRNA splicing, circularization is presumably influenced by both cis-regulatory elements and trans-acting factors, for instance, spliceosome assembly (*Ashwal-Fluss et al., 2014*; *Liang et al., 2017*; *Starke et al., 2015*), topological effects due to intronic sequence (*Chen, 2016*; *Zhang et al., 2014*) and combinatorial effects of RBPs (*Ashwal-Fluss et al., 2014*; *Conn et al., 2015*). Remarkably, the mechanism of back-splicing has been determined to some extent and several factors related to generation of different circRNA isoforms are listed below.

### *Competition of reverse complementary sequences*

Considering that most of circRNAs are generated after their parent genes have been transcribed completely, the majority of circRNA isoforms may occur post-transcriptionally (*Ashwal-Fluss et al., 2014*; *Zhang et al., 2016b*). Reverse complementary sequences flanking the circularized exons, such as abundant Alu elements (*Jeck & Sharpless, 2014*), highly conserved mammalian-wide interspersed repeat (MIR) sequence (*Yoshimoto et al., 2019*) or other non-repetitive complementary sequence (*Memczak et al., 2013*), are efficient to enhance exon circularization by forming paired duplex structures (*Liang & Wilusz, 2014*). Shorter sequences as long as 30 to 40 nucleotides are even able to promote circRNA biogenesis (*Liang & Wilusz, 2014*). Once the intronic complementary sequences across the circularized exons were disrupted, no circRNA could be detected at the examined locus (*Zhang et al., 2016b*).

Theoretically, a series of inverted repeated RNA pairs form different RNA duplexes resulted in multiple circRNA isoforms (Figs. 3B, 3C). Moreover, an individual intron can also form RNA pairing which promotes canonical splicing to linear RNA formation (Fig. 3A). It means that the competition of RNA pairing leads to different splicing products (*Ashwal-Fluss et al., 2014*; *Wilusz, 2018*). Taken together, the endogenous conditions for circRNAs generation are very complex, since the number of repetitive elements, the distance between them, and their degree of complementarity all affect the splicing outcome (*Zhang et al., 2014*).

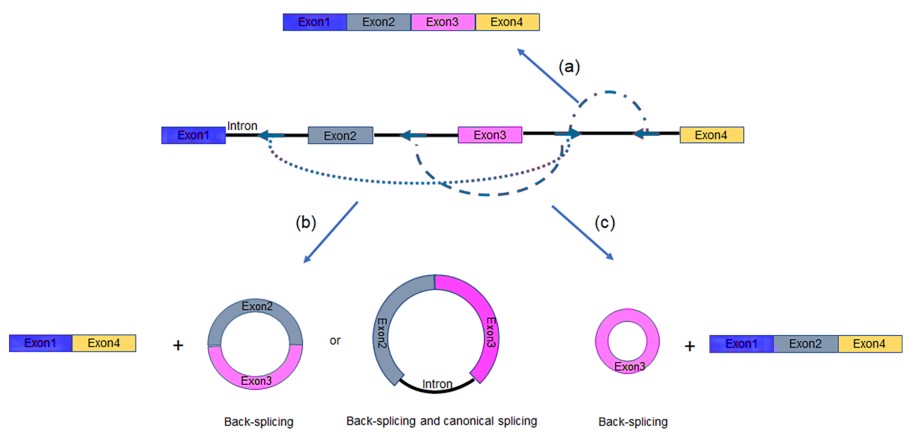

**Figure 3** **Multiple intronic repeats lead to distinct mature RNAs.** (A) An individual intron forms RNA pairing which promotes canonical splicing to linear RNA generation. (B–C) Base pairs flanking the circularized exons results in back-splicing and generation of circRNAs. Blue arrows, intronic repeat elements.

### RNA binding proteins

Various AS events were observed despite of the presumably identical of paired complementary sequences among all tested human samples (*Zhang et al., 2016a*), which indicated that the regulation of AS is more complicated than we found. Previous study showed that the RNA pairing was influenced by RBPs as well (*Ivanov et al., 2015*). These proteins bind to specific intron motifs firstly and then bring the donor site closer to the acceptor site (*Ashwal-Fluss et al., 2014*; *Conn et al., 2015*). In contrast, negative RBPs will suppress circRNA formation by destabilizing RNA pairing interactions, like ADAR1 (*Ivanov et al., 2015*; *Rybak-Wolf et al., 2015*). Differentially expressed RBPs have been reported to mediate pre-mRNA AS in various cell lines, for example, SR and SR-related proteins typically activate splicing (*Nilsen & Graveley 2010*). By contrast, hnRNPs are generally thought to repress splicing. The activity of these proteins can be regulated through signaling networks in specific tissues and cell-types (*McManus & Graveley, 2011*). Hence, we hypothesize that AS in circRNAs may be regulated by similar mechanism.

### Others

It is well known that pre-mRNA processing is closely correlated with polymerase II (Pol II) transcription, and the transcription elongation rate has obvious effect on the occurrence of splicing events (*Bentley, 2014*). Flies with decreased elongation capacity of RNA pol II, produced a significantly lower number of circRNAs (*Ashwal-Fluss et al., 2014*). By applying 4-thiouridine (4sU) to metabolic tagging of nascent RNAs, Zhang et al. discovered that circRNA-producing genes had higher average transcription elongation rate (*Zhang et al., 2016b*). In summary, the positive correlation between nascent circRNA generation and Pol II elongation speed indicates that fast elongation favors RNA folding across flanking introns to form circRNAs rather than within introns (*Bentley, 2014*; *Zhang et al., 2016b*).

Nevertheless, those factors mentioned above still cannot adequately explain the widespread nature of AS in circRNAs. Additional elements that are responsible for

alternative circularization await identification. It is helpful to discover the rules of AS in circRNAs by large-scale identification of full-length circRNAs. Currently, lots of endeavors are focused on the internal structure of circRNAs which further promote their functional characterization and evolutionary analyses.

## Current approaches for identification of alternative splicing

CircRNAs containing unique BSJs, are differ obviously from their host linear RNA counterparts, and hence the BSJs is critical for their identification. So far, a number of biomedical methods have been raised to identify and quantify circRNAs, including RT-PCR/qPCR, Northern blot analyses (*Memczak et al., 2013*; *Zhang, Yang & Chen, 2016*; *Zhang et al., 2013*), and circRNA microarray (*Li et al., 2019*; *Zhang et al., 2018*). Until the development of NGS and bioinformatics tools, the abundant circRNAs were revealed in multiple cell types or tissues.

Many bioinformatic analyzing tools combine all known mRNA exons in a sequential order to represent putative full-length circRNA, which bases on an unsupported assumption that circRNAs possess identical composition with their linear counterparts (*Guo et al., 2014*; *Zhang et al., 2014*). For example, circular(CIRC)explorer (*Zhang et al., 2014*), MapSplice (*Wang et al., 2010*), circRNA_Finder (*Xing & Liu, 2014*), Circular RNA identification (CIRI) (*Gao, Wang & Zhao, 2015*), Find_circ (*Memczak et al., 2013*) and others, are developed for circRNA identification but not assembly of full-length circRNAs.

Given the fact that distinct splicing leads to differential components of exons and/or introns, using BSJs only to represent subsets of circRNA variants greatly limits our understanding of biogenesis, functions and evolution of circRNAs among species. Due to the technically challenging, the aforementioned tools are unable to explore the internal sequence of circRNAs accurately or distinguish AS in linear RNAs from circRNAs derived from the same gene. Recently, a number of efforts have been made to address this challenge.

### CircRNA enrichment methods

Current studies show that circRNAs abundance is approximately ≤10% of their corresponding linear RNA (*Salzman et al., 2013*), and the estimation of circRNA isoforms with lower expression level probably be biased. Therefore, circRNA enrichment is needed prior to RNA library construction and sequencing. Due to the great variability length of circRNAs, it is hard to separate them from other RNA species by size or electrophoretic mobility. Owing to the covalently closed structure, most circRNAs show higher tolerance to Ribonuclease R (RNase R) digestion in comparison with linear RNAs. Nevertheless, utilizing RNase R alone is not sufficient to remove RNAs lacking 3′ overhangs. Therefore, researchers pretreated the RNA samples by combining ribosome RNA (rRNA) depletion with RNase R treatment (*Jeck & Sharpless, 2014*) or depletion of poly(A)+ RNA to ensure that sequencing reads are generated from bona fide circRNAs.

Whereas, some linear RNAs without sufficiently long single-stranded 3′ overhangs, such as small noncoding RNAs, are naturally resistant to RNase R (*Vincent & Deutscher, 2006*). Besides, many polyadenylated mRNAs with complex structures, especially G-quadruplex (G4) structures, are also poor substrates for RNase R, as the enzyme stalls in the body of

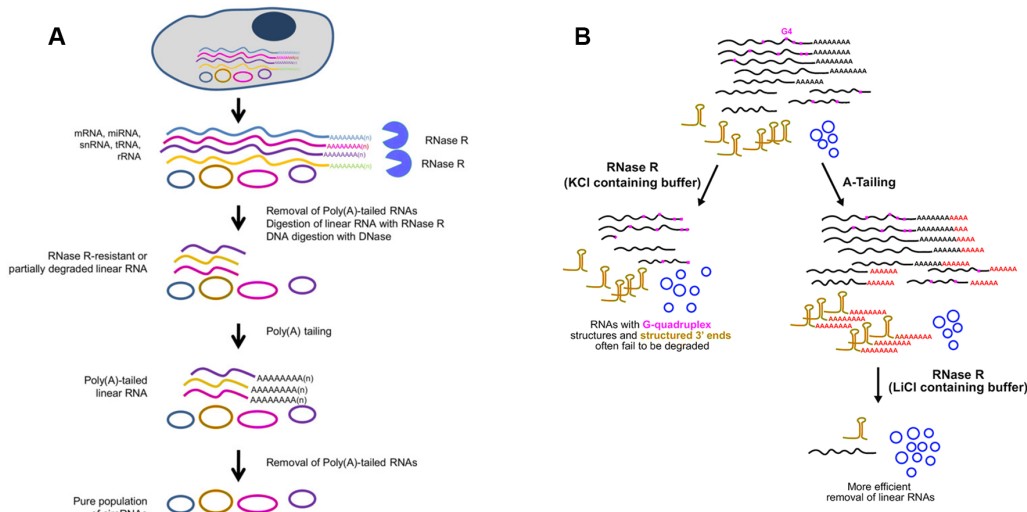

**Figure 4** **Methods used to enrich circRNAs.** (A) RPAD method (*Panda et al., 2017*). Linear RNAs are first depleted by RNase R digestion. After the remaining RNA is polyadenylated, a second round of depletion of poly(A)+ RNAs using oligo(dT) beads leaves a highly enriched population of circular RNAs. (B)Coupling A-Tailing with RNase R treatment in LiCl containing buffer (*Xiao & Wilusz, 2019*). Total RNA is treated with RNase R in KCl-containing buffer (left). Total RNA is treated with an A-Tailing step followed by RNase R digestion in LiCl-containing buffer (right). Compared with the original image, the red/green colors were adjusted.

these transcripts (*Panda et al., 2017*; *Xiao & Wilusz, 2019*). Considering that the expression of circRNAs is often lower than the linear counterparts, even a small amount of undigested linear RNAs may surpass the abundant of cognate circRNAs (*Panda et al., 2017*). To resolve this major obstacle, Panda et al. raised another approach to isolate high-purity circRNA termed RNase R treatment followed by Polyadenylation and poly(A)+ RNA Depletion (RPAD) (Fig. 4A) (*Panda et al., 2017*; *Pandey et al., 2019*). After RNase R treatment, small RNAs were decreased to 10–50% of the remaining by polyadenylation and poly(A)+ RNA depletion. In an alternative manner, Xiao et al. found that RNase R could digest linear RNAs more efficiently by using Li+ instead of K+ (which stabilizes G4s) in the reaction buffer (*Xiao & Wilusz, 2019*). Drawing inspiration from the RPAD method, purer circRNAs can be isolated by coupling A-tailing with RNase R in LiCl-containing buffer (Fig. 4B) (*Xiao & Wilusz, 2019*).

Compared with conventional models of circRNA enrichment using rRNA depletion kits of model organisms, the other two methods can be used in non-model organisms for efficient removal of linear RNAs regardless of the RNA sequences. A noteworthy drawback of RPAD is that quantification of circRNA abundance may not be strictly accurate due to involvement of many enzymatic and RNA isolation steps. Furthermore, according to specific experiment purpose, RPAD protocol was modified via adding rRNA depletion step before RNase R treatment (*Rahimi et al., 2019*). It is worth noticing that extensive purification steps will cause significant biases on circRNA abundance.

**Table 1  Recently published bioinformatic tools used for AS detection in circRNAs.**

| Algorithm | Sample | Sequencing platform | Application and limitations | Reference |
|---|---|---|---|---|
| CIRI-AS | RNase R- or RNase R+ treated samples | Illumina HiSeq 2500 | Detects AS within circRNAs but has lower sensitivity for short reads | *Gao et al. (2016)* |
| FUCHS | rRNA- or rRNA-/RNase R+ treated samples) | Illumina MiSeq system or Illumina HiSeq2500 | Identifies alternative spliced circles and visualizes the coverage profile of circRNAs | *Metge et al. (2017)* |
| CIRCexplorer2 | Poly(A)+, poly(A)- and/or RNase R-treated samples | Illumina HiSeq 2000 | Detects alternative (back-) splicing circRNAs and parallel poly(A)+ RNA-seq is needed | *Zhang et al. (2016a)* |
| CircSplicer | RNA-/RNase R-treated samples | Illumine Hiseq 2500 | AS detection between cancer and normal conditions | *Feng et al. (2019)* |
| circseq_cup | rRNA-/RNase R-treated samples | Illumina Hiseq 3000 | Assembly of full-length circRNAs | *Ye et al. (2017)* |
| CircAST | rRNA-/RNase R-treated samples | Illumina Hiseq 1500 | Assembles and quantifies exonic circRNA isoforms but may miss intronic or intergenic circRNAs | *Wu et al. (2019)* |
| CIRI-full | rRNA-/RNase R-treated samples without fragmentation | Illumina HiSeq 2500 (PE250 or PE300) | Reconstructs and quantifies circRNA isoforms by utilizing longer RNA-seq data | *Zheng et al. (2019)* |

**Notes.**
Abbreviations: rRNA, ribosomal RNA depletion; poly(A)+, polyadenylated RNA; poly(A)-, non-polyadenylated RNA; PE, paired-end.

### Genome-wide annotation of full-length circRNAs

At present, a combination of RNA-seq analysis with bioinformatic survey has been widely used for large-scale determination of circRNAs. Moreover, various algorithms have been raised to analyse sequencing information generated from different platforms.

*Next-generation sequencing methods.* Studies have characterized circular variants arising from one host-gene with ABSJs sites (*Jeck et al., 2013*; *Zhang et al., 2014*), and these isoforms can be identified by existing circRNA detection algorithms (*Peng et al., 2020*). In order to identify circRNA isoforms with the same BSJs, more efficient and stringent tool have been developed and listed in Table 1.

 CIRI-AS has been designed to investigate internal components of circRNAs for the first time (*Gao et al., 2016*). This algorithm is based on reconstructing circRNA exons (cirexons) routes and clustering alternatively spliced cirexons, and could be applied to majority of current available RNA-seq data. Likewise, FUCHS focuses on the AS events within same circle boundaries by analyzing long sequencing reads (typically >150 bp) (*Metge et al., 2017*). In an alternative manner, CIRCexplorer2 predicts AS in circRNAs by comparing data sets between poly(A)+ and poly(A)- RNA-seq (*Zhang et al., 2016a*). Compared with CIRI-AS, CircSplice could detect more AS events and provide the comparison function of different samples (*Feng et al., 2019*). According to *Ye et al. (2017)* a bioinformatics pipeline named circseq_cup was developed to assemble full-length circRNAs, which utilized fusion junction sites and their corresponding paired-end RNA-seq reads. The length of circRNAs identified by this approach relied on the read length and sequencing library length. As

for CircAST, it is developed for reconstruction and quantification of circRNA variants. Additionally, it shows better performance on variable read lengths (from 75 bp to 125 bp). However, as an annotated-based method, CircAST may miss intronic or intergenic circRNAs (*Wu et al., 2019*). CIRI-full reconstructs full-length circRNAs by combining BSJ and reverse overlap (RO) features and facilitates the identification of low-abundance circRNAs. Meanwhile, it is more suitable for longer sequencing reads (>250 or 300 bp) (*Zheng et al., 2019*). Compared with previous methods, the main advantage is that CIRI-full uses unfragmented RNA samples for library preparation.

In summary, these tools have different advantages in detection of circRNA isoforms. Most notably, CIRI-full provides precise sequence of circRNAs ($\leq$ 600 bp) by using cDNA libraries without fragmentation step. On the contrary, the majority of algorithms provides indirect data of circRNAs but perform well even using short read sequencing data. However, they all face an inherent challenge that the reconstruction of circRNAs relies on the read length as well as the insertion sizes of cDNA libraries (*Ye et al., 2017*), so they are limited to small circRNAs and biased on prediction of large circRNAs (*Gao et al., 2016*).

*Long-read sequencing methods.* According to the aforementioned that read length is the key determinant of full-length circRNAs validation, the utilization of third-generation sequencing (TGS) technologies, such as PacBio long read sequencing, is likely to identify circRNA variants accurately. By using Oxford Nanopore Technology (ONT), Rahimi et al. performed global examination of full-length circRNAs in human and mouse brains (*Rahimi et al., 2019*). According to this study, the enriched circRNA pool was linearized and re-polyadenylated prior to library preparation. This study, to some degree, circumvents the limitation of NGS methods and provides a fast and reliable method of circRNA reconstruction. However, its greatest drawback is that fragmentation of circRNAs will probably influence the accuracy of subsequent assembly even though the linearization process has been optimized.

### Experimental validation of full-length circRNAs by rolling circle amplification

Compared with bioinformatics reconstruction, experimental approaches provide precise sequence of specific circRNAs most intuitively. Hence, follow-up experimental validation for assembled circRNAs may then be applied to further strengthen the case for existence of many circRNAs. As reverse transcriptase has potential strand displacement activity (*Kelleher, 1998*), it allows the displacement of any complementary sequence hybridized downstream to produce many copies of the same template (*Acevedo, Brodsky & Andino, 2014*).

Owing to the unique structure of circRNAs, multiple rounds of reverse transcription could produce double or triple sized products (*Barrett, Wang & Salzman, 2015*; *Danan et al., 2012*; *Starostina et al., 2004*). *You et al. (2015)* for the first time, detected the full-length of circRNAs by sequencing of rolling circle cDNA products on PacBio platform. Through creating circular-derived PCR products with divergent primers, Hirsch et al. combined ONT sequencing with a PCR-based approach to gain insight into the internal structure of circNPM1 (*Hirsch et al., 2017*). Moreover, Das et al. presented a novel method,

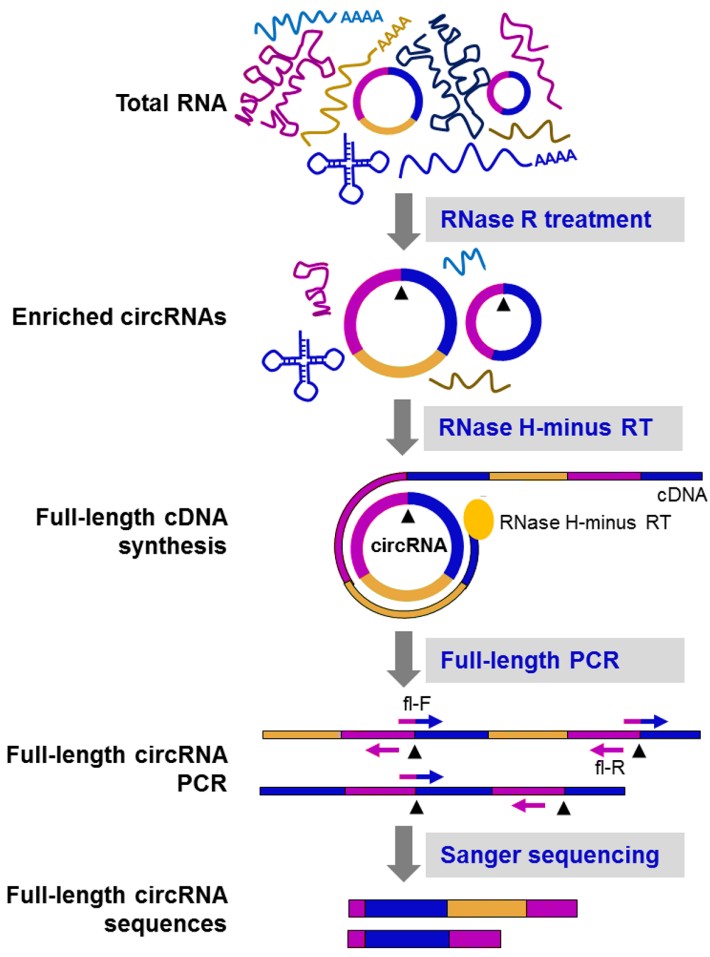

**Figure 5   CircRNA-Rolling Circle Amplification (circRNA-RCA) method enables identification of circRNA variants (*Das et al., 2019*).** Firstly, RNase R is used to enrich the circRNA. Then, the enriched circRNA is reverse-transcribed with RNase H-minus reverse transcriptase and cDNA is amplified by PCR, using full-length primers. Finally, the full-length circRNA PCR products are sequenced by the Sanger method to identify circRNA variants. Black arrowheads: back-splice site. Compared with the original image, the red/green colors were adjusted.

named circRNA-Rolling Circle Amplification (circRNA-RCA), aiming to distinguish circRNA variants containing the same BSJ site but distinct in internal sequence (Fig. 5). In this method, a forward primer spanning the BSJ sequence and a reverse primer exactly upstream of the forward primer were used to produce tandem-repeat cDNA amplicons with the supplementation of RNase H-minus reverse transcriptase (*Das et al., 2019*). However, the main drawback of these methods is that only a handful of circRNAs can be examined and the sensitivity and /or specificity is not satisfactory.

## CONCLUSIONS

Recent advances have revealed partial factors related to the AS events within circRNAs. However, the regulation of AS in endogenous conditions is far more complex than we

have discovered and requires further investigation. To some extent, identification of their full-length sequence contributes to further understanding the regulation and function of AS. In light of the issues, novel bioinformatics tools have been raised to reconstruct circRNA isoforms such as CIRI-AS and CircAST. However, they provide indirect data of circRNAs due to the fragmentation step before library preparation. Despite the utilization of non-fragmented samples, CIRI-full is difficult to completely recover large circRNAs which is limited by the short-read length of NGS. Remarkably, experimental approaches like circRNA-RCA provide precise sequence of specific circRNAs, while only a limited subset of circRNAs could be validated.

In summary, there is no competent method that is applicable for large-scale circRNAs detection with high accuracy. We believe that the combination of rolling circle amplification with TGS technologies is a promising method for full-length circRNAs identification. And with the development of methodologies, new insights will be provided on understanding roles of circRNAs and previous unknown biological phenomena in the coming years.

### Funding
This work was funded by the National Natural Science Foundation of China (grant number 61871121). The funders had no role in study design, data collection and analysis, decision to publish, or preparation of the manuscript.

### Grant Disclosures
The following grant information was disclosed by the authors:
National Natural Science Foundation of China: 61871121.

### Competing Interests
The authors declare there are no competing interests.

### Author Contributions
- Xiaohan Li conceived and designed the experiments, performed the experiments, analyzed the data, prepared figures and/or tables, authored or reviewed drafts of the paper, and approved the final draft.
- Bing Zhang performed the experiments, authored or reviewed drafts of the paper, and approved the final draft.
- Fuyu Li and Kequan Yu analyzed the data, authored or reviewed drafts of the paper, and approved the final draft.
- Yunfei Bai conceived and designed the experiments, authored or reviewed drafts of the paper, and approved the final draft.

### Data Availability
There was no raw data or code used in this review.

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
