# Peer review of "The mechanism and detection of alternative splicing events in circular RNAs"

_PeerJ, doi:10.7717/peerj.10032_

## Round 0.1 · original submission · Minor Revisions

Dear Dr. Li and colleagues:

Thanks for submitting your manuscript to PeerJ. I have now received three independent reviews of your work, and as you will see, the reviewers raised some minor concerns about the manuscript. Despite this, these reviewers are optimistic about your work and the potential impact it will have on research studying mechanisms and detection of alternative splicing events in circular RNAs. Thus, I encourage you to revise your manuscript, accordingly, taking into account all of the concerns raised by both reviewers.

There are not too many suggestions; thus, it should not take much effort to address these concerns to greatly improve your manuscript.

Please have an English-speaking expert help revise your manuscript.

Please note that reviewers 2 and 3 have included marked-up versions of your manuscript.

I look forward to seeing your revision, and thanks again for submitting your work to PeerJ.

Good luck with your revision,

-joe

·

Basic reporting

The review is well structured, however I recommend that a native English speaker is consulted as there are several occasions where a wrong word is used. The review shows a comprehensive and adequate literature review. The review might be interesting for scientists studying alternative splicing within circRNAs however there has been a recent review paper in Briefings in Bioinformatics (Chen et al. Feb. 2020: The bioinformatics toolbox for circRNA discovery and analysis) summarizing even more tools and resources for circRNAs. Li at al has a slightly stronger focus on alternative splicing and mentions only two programs Chen et al. do not mention.

Experimental design

The review mentions many programs for alternative splicing in circRNAs, however it would be good if they had also compared advantages and disadvantages of the programs or tried to run selected programs to give the reader advice on which tool to use for which type of data.

Validity of the findings

Li et al. nicely highlights the drawbacks of short read sequencing when trying to reconstruct full length circRNAs.

Additional comments

The review nicely summarizes the current field and might be a good starting point for further reading.

·

Basic reporting

The authors have compiled an informative and insightful review of the mechanism and detection of alternative splicing events in circular RNAs. This reviewer found the topic timely and of great interest, and the article informative. Overall, I recommend this manuscript for publication pending minor edits.

Experimental design

'no comment'

Validity of the findings

'no comment'

Additional comments

A few suggestions would help to provide a bit more in-depth detail and answer questions that naturally arise from the article's discussion. Also, moderate English editing would be helpful to clarify the point of several sentences.

·

Basic reporting

The present manuscript submitted by the authors on circRNA is considered suitable for publication pending major edits. Despite presenting a fairly complete content regarding the topic it develops, it is recommended to detail a little more information, as well as a review of the English language. However, I consider it is important to emphasize that they bring very interesting information to the field, giving enough literature references and an adequate article structure.

Experimental design

The article presented complies with PeerJ's aim & scopus and the information they provide is considered sufficient and appropriate. However, some claims are requested to be discussed and more detailed.

Validity of the findings

Despite not being a research article, they review interesting information for the circRNA field, providing concise information on the subject.

Additional comments

The present manuscript submitted by the authors on circRNA is considered suitable for publication pending major edits. Despite presenting a fairly complete content regarding the topic it develops, it is recommended to detail a little more information listed below, as well as a review of the English language. However, I consider it is important to emphasize that they bring very interesting information to the field.

---

## Round 0.2 · accepted · Accept

Dear Dr. Li and colleagues:

Thanks for revising your manuscript based on the concerns raised by the reviewers. I now believe that your manuscript is suitable for publication. Congratulations! I look forward to seeing this work in print, and I anticipate it being an important resource for groups studying mechanisms and detection of alternative splicing events in circular RNAs. Thanks again for choosing PeerJ to publish such important work.

Best,

-joe

·

Basic reporting

no comment

Experimental design

no comment

Validity of the findings

no comment

·

Basic reporting

No comment

Experimental design

No comment

Validity of the findings

No comment

Additional comments

The authors have adequately answered all the queries proposed in the previous review. I believe that figures, with the modifications that they made and the greater explanation of them in the figure caption, are much better understood. Congratulations for your job.